# Pred-SF: A Precipitation Prediction Model Based on Deep Neural Networks

**DOI:** 10.3390/s23052609

**Published:** 2023-02-27

**Authors:** Rongnian Tang, Pu Zhang, Jingjin Wu, Youlong Chen, Lingyu Dong, Song Tang, Chuang Li

**Affiliations:** 1Electrical and Mechanical College, Hainan University, Haikou 570228, China; 2Meteorological Information Center of Hainan Province, Haikou 570203, China; 3Key Laboratory of South China Sea Meteorological Disaster Prevention and Mitigation of Hainan Province, Haikou 570106, China

**Keywords:** predict precipitation, Pred-SF, self-cyclic, step by step, PredRNN-V2, ERA5, GPM

## Abstract

How to predict precipitation accurately and efficiently is the key and difficult problem in the field of weather forecasting. At present, we can obtain accurate meteorological data through many high-precision weather sensors and use them to forecast precipitation. However, the common numerical weather forecasting methods and radar echo extrapolation methods have insurmountable defects. Based on some common characteristics of meteorological data, this paper proposes a Pred-SF model for precipitation prediction in target areas. The model focuses on the combination of multiple meteorological modal data to carry out self-cyclic prediction and a step-by-step prediction structure. The model divides the precipitation prediction into two steps. In the first step, the spatial encoding structure and PredRNN-V2 network are used to construct the autoregressive spatio-temporal prediction network for the multi-modal data, and the preliminary predicted value of the multi-modal data is generated frame by frame. In the second step, the spatial information fusion network is used to further extract and fuse the spatial characteristics of the preliminary predicted value and, finally, output the predicted precipitation value of the target region. In this paper, ERA5 multi-meteorological mode data and GPM precipitation measurement data are used for testing to predict the continuous precipitation of a specific area for 4 h. The experimental results show that Pred-SF has strong precipitation prediction ability. Some comparative experiments were also set up for comparison to demonstrate the advantages of the combined prediction method of multi-modal data and the stepwise prediction method of Pred-SF.

## 1. Introduction

Precipitation is one of the most common weather phenomena affecting social production and human life. Accurately predicting the change of precipitation can improve people’s production and work efficiency, take protective measures before the occurrence of extreme weather such as a rainstorm, ensure people’s life safety and reduce social economic losses.

With the arrival and rapid development of the information age, more and more meteorological data can be obtained by a high-precision meteorological measurement sensor, which provides the data basis for numerical weather prediction [1] to forecast the weather objectively and quantitatively. This method has the ability for long-time modeling of meteorological change and a strong physical interpretation ability, so it has been the mainstream precipitation forecasting method. However, the calculation cost is high, and the physical equations used are deduced by scientists from the collected meteorological data, which cannot represent the complete and real changing state of the weather system. In addition, the radar echo extrapolation method [2] has also become one of the mainstream methods in the field of precipitation prediction because of its excellent prediction ability of short-term severe convective weather. The commonly used radar echo extrapolation methods include the centroid method [3], cross correlation method [4] and optical flow method [5].

With the rise of deep learning, the meteorological field has a new development direction, and more and more researchers apply deep learning methods to precipitation prediction. Among them, the convolutional neural network is widely used in the field of precipitation forecast because of its excellent spatial data information extraction and fusion capability [6,7,8,9]. On the other hand, the long- and short-term memory network is favored by meteorologists for its excellent ability to build a long-distance contextual data dependence model [10,11,12]. In addition, the generative adversarial network has strong image generation ability and also occupies a place in the field of precipitation prediction [13,14].

To further explore the problem of precipitation prediction in the spatial-temporal dimension, Shi et al. proposed the ConvLSTM network [15], which replaced the fully connected structure in LSTM with the convolution structure, and used this network to obtain better results than the optical flow method on the radar echo dataset. With the ConvLSTM, the use of a spatial-temporal series model to solve the problem of precipitation forecast began to enter a rapid development stage. Shi et al. also proposed the TrajGRU network [16], and associated the problem of spatial-temporal series prediction with the problem of precipitation prediction. This model achieved better results than ConvLSTM on the radar echo dataset. Wang et al. added spatio-temporal memory cell M into ConvLSTM’s network unit and proposed a PredRNN model with better spatio-temporal prediction ability [17]. Subsequently, they also made many innovations in the unit structure and overall framework and successively proposed PredRNN++ [18], MIM [19], E3D-LSTM [20], PredRNN-V2 [21] and other models, which have achieved great results in radar echo prediction of precipitation or other spatio-temporal prediction problems. Le Guen et al. proposed a new recursive unit PhyCell, used partial differential equations as a dynamic supplement to model prediction, and proposed a PhyDNet model [22], which promoted the development of spatio-temporal prediction models.

The continuous introduction of these spatio-temporal prediction models has promoted the application of deep learning models in the precipitation forecast business. However, these models have only been tested and applied on the radar echo dataset, and there will be large prediction errors when the meteorological system is not stationary. Moreover, the input and output tensors of these models are the same size, which means that multi-frame images with H×W length and width are used to predict the image changes in the same range. When applied to the precipitation prediction problem, the precipitation situation at the edge of the target region is difficult to predict.

To obtain long-term stable precipitation prediction results, this paper proposed a two-step model Pred-SF to predict precipitation in target areas. This model respects the laws of meteorological physical changes and uses the multi-modal data of $T_{i}$ moments in the history of a large area centered on the target area (referred to as the large area) to predict the precipitation values of the subsequent $T_{o}$ moments in the target area. Pred-SF divides the prediction of precipitation in target areas into two steps. In the first step, a spatial encoding structure and PredRNN-V2 autoregressive spatio-temporal prediction network are selected to extract and fuse spatio-temporal features of multi-modal meteorological data in a large area, including precipitation. In this process, a preliminary prediction of multi-modal meteorological data is made. The second step is to use the spatial information fusion network to further integrate the spatial characteristics of the multi-meteorological modal data output from the first part, so that the model focuses on the prediction and output of the precipitation in the target region. We used ERA5 and GPM data to test the model, and the experimental results show that the multi-step prediction method of Pred-SF and the combined prediction method of multi-modal data have great advantages.

## 2. Data

### 2.1. Analysis of Meteorological Data

In the problem of meteorological prediction, the precipitation change in the target area is complex and dynamic and continuous, and it is also the result of the comprehensive action of various meteorological modal data in the target area and surrounding area. After observing a large amount of meteorological data, we can get some basic data characteristics. Meteorological data have four characteristics, namely time-series continuity, time-series periodicity, spatial feature diversity and spatial regional correlation.

The continuity of time series means that meteorological data change continuously over time, and there is some correlation between meteorological values at different moments. The meteorological values at a certain moment will be affected by the meteorological values at historical moments. The periodicity of time series means that much meteorological data follow periodic changes on the whole [23]; for example, temperature changes continuously in daily and yearly cycles. Spatial feature diversity refers to the correlation between various meteorological modes in a specific spatial region. The generation of a certain meteorological phenomenon in a spatial region is often the result of the combined action of various meteorological modes in the region, rather than only determined by one meteorological datum. Spatial regional correlation refers to the correlation between meteorological data of different spatial regions. Meteorological data of a certain spatial region are often related to that of its neighboring regions; especially in the process of dynamic changes of meteorological data, this correlation is more obvious.

Based on the characteristics of these four kinds of meteorological data, Pred-SF has made corresponding adjustments in the selection and processing of input data. The input of Pred-SF is continuous time-series meteorological data, and time latitude including month, day and hour is added to the input data to satisfy periodicity. For precipitation prediction, the model mainly uses the multi-modal meteorological data including precipitation as the input data of the model, in which precipitation is the main predicted value, and the multi-modal data are used to simulate the real meteorological environment. At the same time, the model adds the large-scale regional spatio-temporal data with the target region as the center to predict the spatio-temporal data of the target region. Using a large range to predict a small range can better capture the temporal and spatial characteristics of precipitation in the target region during the dynamic change of the meteorological system.

### 2.2. Dataset and Study Area

In this study, we used GPM and ERA5 meteorological data. GPM is the global precipitation measurement data obtained by NASA through the integrated multi-satellite inversion technology (IMERG) [24]. GPM data are grid data with a spatial resolution of 0.1 longitude times 0.1 latitude and a time resolution of 30 min. In this study, we approximated the GPM data as the real precipitation value. ERA5 is a new generation of global climate and atmosphere reanalysis data developed by the European Center for Medium-Range Weather Forecasts [25]. ERA5 data are also grid data, with a variety of meteorological modes including temperature, humidity, wind speed and other commonly used data, and have a spatial resolution of 0.25 longitude times 0.25 latitude and a temporal resolution of one hour. In this study, we chose the temperature, U wind, V wind, vertical wind, specific humidity, cloud cover and percentage of cloud liquid water content of seven kinds of meteorological modals, and select the mode in the 500 hpa, 700 hpa, 850 hpa, 950 hpa and 1000 hpa five vertical-pressure-layer data. We use these data in conjunction with GPM precipitation measurements to model real climate conditions. We have listed the specific data information in Table 1.

Our precipitation forecast target area is 109.8 to 116.2 longitude and 19.8 to 26.2 latitude, expressed as a 64 × 64 grid area using GPM data. However, because the precipitation region changes dynamically with time, to obtain the precipitation prediction results of the target region in the following period of time, we added 128 × 128 GPM grid data, which is centered on the target region. A large range of precipitation data is used as a dynamic supplement to better predict the precipitation changes at the edge of the target area. To simulate the climate environment of this region, we selected ERA5 data covering the target prediction area, ranging from 109 to 117 longitude and 19 to 27 latitude, and the ERA5 data can be represented as a 32 × 32 grid area. We have given the study area in Figure 1.

## 3. Preliminaries

### 3.1. Spatio-Temporal Prediction and Precipitation Prediction

Spatio-temporal sequence data refer to the data embedded in a fixed space with both temporal and spatial correlation; that is, the temporal dimension and spatial location are not independent of each other, but interrelated. The usual expression of spatio-temporal sequence data is $X\in R^{T\times C\times H\times W}$, where T is a continuous time series, C is the number of data channels and H and W are used to represent the spatial size of the data. The model that uses the spatio-temporal series data of the historical continuous time to predict the spatio-temporal series data of the future continuous time is called the spatio-temporal prediction model. 

The continuous meteorological data in a fixed spatial region is a kind of spatio-temporal sequence data, which can also be represented by $X\in R^{T\times C\times H\times W}$, where T represents the continuous meteorological data series, C represents the types of meteorological modes in the space and H and W represent the spatial size of the region where the meteorological data are located. Therefore, the meteorological data prediction problem in the field of meteorology can be regarded as a spatio-temporal prediction problem, and the precipitation prediction problem, as a classical meteorological data prediction problem, can also be regarded as a spatio-temporal prediction problem, which can be solved by the spatio-temporal prediction model. 

### 3.2. Spatio-Temporal Prediction Model PredRNN and PredRNN-V2

In the field of deep learning, autoregressive prediction is a commonly used spatio-temporal prediction network framework, which can predict the spatio-temporal information of multiple successive time steps. The main structure is a cyclic prediction model that stacks multi-layer LSTM network units or variants and other modules, such as PredRNN, MIM, PhyDNet, etc. 

Based on the ConvLSTM network, Wang et al. innovated the structure and network units and proposed the PredRNN model, which is an autoregressive spatio-temporal prediction model. The temporal and spatial memory of the network exists not only in the LSTM network units, but also in the stacked structure of the model. Aiming at the problem that the spatio-temporal information of the topmost network unit at time t-1 will be ignored by the lowest network unit at time t, the network transmits the information extracted from the last network unit at each time to the first layer at the next time to enhance the overall spatio-temporal information memory ability of the model. In the network unit, in addition to the original time memory cell C, PredRNN also adds the space-time memory cell M. The temporal memory cell C is used for the lateral connection of the model, and its main function is to remember the spatial information at the previous moment. The spatio-temporal memory cell M is used for the longitudinal connection of the multi-layer stacking model, and the M state of the last layer of network unit at each time will be transmitted to the first layer at the next time. Its main function is to remember the spatial information of the previous layer and the last layer at the previous time. The overall space-time information flow equation is as follows:(1)$$g_{t}=\tanh\left(W_{xg}*X_{t}+W_{hg}*H_{t-1}^{l}+b_{g}\right)\;i_{t}=\sigma \left(W_{xi}*X_{t}+W_{hi}*H_{t-1}^{l}+b_{i}\right)\;f_{t}=\sigma \left(W_{xf}*X_{t}+W_{hf}*H_{t-1}^{l}+b_{f}\right)\;C_{t}^{l}=f_{t}⊙C_{t-1}^{l}+i_{t}⊙g_{t}\;{g_{t}}^{\prime }=\tanh\left({W_{xg}}^{\prime }*X_{t}+W_{mg}*M_{t}^{l-1}+{b_{g}}^{\prime }\right)\;{i_{t}}^{\prime }=\sigma \left({W_{xi}}^{\prime }*X_{t}+W_{mi}*M_{t}^{l-1}+{b_{i}}^{\prime }\right)\;{f_{t}}^{\prime }=\sigma \left({W_{xf}}^{\prime }*X_{t}+W_{mf}*M_{t}^{l-1}+{b_{f}}^{\prime }\right)\;M_{t}^{l}={f_{t}}^{\prime }⊙M_{t}^{l-1}+{i_{t}}^{\prime }⊙{g_{t}}^{\prime }\;O_{t}=\sigma \left(W_{xo}*X_{t}+W_{ho}*H_{t-1}^{l}+W_{co}*C_{t}^{l}+W_{mo}*M_{t}^{l}+b_{o}\right)\;H_{t}^{l}=o_{t}⊙\tanh\left(W_{1\times 1}*\left[C_{t}^{l},\;M_{t}^{l}\right]\right).$$

The prediction performance of the proposed network is better than that of ConvLSTM on a mobile handwritten digit dataset and radar echo dataset. 

PredRNN model has transverse cross-time extension path and longitudinal spatial feature fusion path, which enables the network to learn complex nonlinear transition function of short-term motion. However, this deep temporal structure may also bring a gradient disappearance problem. To solve this problem, Wang et al. proposed the PredRNN-V2 model, which combines the temporal memory cell C and the spatio-temporal memory cell M together with a decoupled structure, so that the model can make longer-term predictions. 

We take PredRNN-V2 combined with a spatial encoding structure as the backbone of the first part of the Pred-SF model to jointly forecast multi-modal meteorological data in multi-time steps.

## 4. Methods

### 4.1. Overview

We regard the precipitation prediction problem in the field of meteorology as a spatio-temporal prediction problem in the field of deep learning and design the Pred-SF weather prediction model, as shown in Figure 2.

The whole model can be expressed as:(2)$$\left({\tilde{D}}_{P}{}^{T+1},\;{\tilde{D}}_{P}{}^{T+2},\;\ldots ,\;{\tilde{D}}_{P}{}^{T+T_{o}}\right)=Pred-SF\left(D^{T},\;D^{T-1},\;\ldots ,\;D^{T-T_{i}+1}\right)$$
where D represents the input data of the model, $D^{t}\in R^{C_{i}\times T_{i}\times H_{i}\times W_{i}}$. $C_{i}$ represents the number of channels for the input multi-modal meteorological data. $T_{t}$ represents the time step of the input data. $H_{i}$ and $W_{i}$ represent the spatial size of the input data. ${\tilde{D}}_{P}$ means the output data of the model, ${\tilde{D}}_{P}\in R^{C_{o}\times T_{o}\times H_{o}\times W_{o}}$. $C_{o}$ means the type of meteorological data (precipitation in this model). $H_{o}$ and $W_{o}$ mean the spatial dimensions of the output data. 

Pred first extracts and fuses the spatio-temporal features of the input multimodal data and outputs the predicted value of the multimodal data of the same region and type.
(3)$$\left({\tilde{D}}^{T+1},\;{\tilde{D}}^{T+2},\;\ldots ,\;{\tilde{D}}^{T+T_{o}}\right)=Pred\left(D^{T},\;D^{T-1},\;\ldots ,\;D^{T-T_{i}+1}\right)$$
where $\tilde{D}\;$ represents the output data of PredRNN-V2, $\tilde{D}\in R^{C_{i}\times T_{o}\times H_{i}\times W_{i}}$, according to the characteristics of the autoregressive space-time prediction model; $\tilde{D}$ and D have the same number of channels and size of data space, all of which are $C_{i}$ and $H_{i}\times W_{i}$. 

Based on the output value of Pred, SF further extracts and fuses the spatial features of multi-modal data, and finally outputs the predicted precipitation value of the target region.
(4)$$\left({\tilde{D}}_{P}{}^{T+1},\;{\tilde{D}}_{P}{}^{T+2},\;\ldots ,\;{\tilde{D}}_{P}{}^{T+T_{o}}\right)=SF\left({\tilde{D}}^{T+1},\;{\tilde{D}}^{T+2},\;\ldots ,\;{\tilde{D}}^{T+T_{o}}\right)$$

Ensuring that the model can be predicted in two steps is achieved by calculating two losses. In the process of model training, the predicted value $\left({\tilde{D}}^{T+1},\;{\tilde{D}}^{T+2},\;\ldots ,\;{\tilde{D}}^{T+T_{o}}\right)$ of Pred needs to calculate the first loss with the real multi-meteorological modal data $\left(D^{T+1},\;D^{T+2},\;\ldots ,\;D^{T+T_{o}}\right)$, and the predicted value $\left({\tilde{D}}_{P}{}^{T+1},\;{\tilde{D}}_{P}{}^{T+2},\;\ldots ,\;{\tilde{D}}_{P}{}^{T+T_{o}}\right)$ of SF needs to calculate the second loss with the real precipitation value $\left(D_{P}{}^{T+1},\;D_{P}{}^{T+2},\;\ldots ,\;D_{P}{}^{T+T_{o}}\right)$ of the target area (where $D_{P}$ represents the real precipitation value in the target region). The two losses are added to the final loss value of the model. The model will reduce the loss value through the gradient descent algorithm and optimize its own parameters to make the model predict step by step.

### 4.2. Spatio-Temporal Prediction of Multimodal Data

As the first step of the model, Pred contains the feature coding structure and the PredRNN-V2 network. In this part, we mainly establish the longitudinal cross-spatial dimension information fusion path and the horizontal cross-temporal dimension information memory path. After the input of the multi-modal data of $T_{i}$ time steps, we can simulate the near-real weather system changes frame by frame. In this process, the multi-modal data prediction values of the same region and type in the following $T_{o}$ time steps are obtained.

The function of the feature coding structure is to preliminarily fuse the spatial information of different meteorological data and initially establish the spatial relationship between different meteorological data. The feature coding structure mainly uses a two-layer convolutional neural network to process the input data. The mechanism of the convolutional neural network is to use multiple different filters to carry out convolution operations on all channels of the three-dimensional tensor in the form of $R^{C_{1}\times H_{1}\times W_{1}}$ and regenerate the three-dimensional tensor in the form of $R^{C_{2}\times H_{2}\times W_{2}}$. In this process, the values between different channels will be multiplied by the convolution kernel and then added, and the spatial feature extraction and fusion of multi-modal data also take place in this process.

The main function of the PredRNN-V2 network is to provide an autoregressive cyclic prediction structure. The spatial encoding structure first inputs the encoded data at time t into the PredRNN-V2 network, and PredRNN-V2 will output the multimodal data prediction result ${\tilde{D}}^{t+1}$. At the same time, this prediction result will be used as input to predict the subsequent multi-modal data ${\tilde{D}}^{t+2}$, thus forming a frame-by-frame circular prediction.

In the process of Pred, different kinds of meteorological data and the same kind of meteorological data are continuously integrating and reconstructing temporal and spatial information, which makes each kind of meteorological datum predicted at each step the result of comprehensive calculation of multiple meteorological data at historical moments before time t, improving the overall prediction accuracy of multi-modal data. On this basis, the prediction accuracy of precipitation is improved.

### 4.3. Spatial Information Fusion

SF is the second step of the model, which mainly uses the multi-layer convolutional neural network to carry out spatial information fusion on the output results of Pred, so as to obtain more accurate precipitation prediction results in the target region and take the results as the final output of the model.

In this step, the input data are the multi-modal forecast value ${\tilde{D}}^{t}$, which contains the crude forecast value of precipitation. To further improve the prediction accuracy of precipitation in the target region, we need to carry out targeted precise prediction on the basis of the crude prediction, that is, to extract and fuse the spatial features of ${\tilde{D}}^{t}$ of multi-modal data at this moment through a multi-layer convolution operation. Output the predicted precipitation value ${\tilde{D}}_{P}{}^{t}$ of the target region. Thus, the predicted ${\tilde{D}}_{P}{}^{t}$ not only contains the spatio-temporal information of various meteorological data at the historical time, but also the spatial information of various meteorological data at the time t.

### 4.4. Training

This paper verified the effectiveness of the model by predicting the subsequent 4 h continuous precipitation in the target area.

The prediction label of the model is the real precipitation in the target area for 4 consecutive hours after a certain time, namely, the GPM precipitation data. The input data of the model were ERA5 multi-modal data in a large area for 4 h in a historical period, temporal dimension data, GPM data in the target area and GPM data in a large area. Because the resolution of GPM data is 30 min, the GPM data half an hour before each time is also added to the input data at that time to increase more spatial and temporal characteristics of precipitation and improve the prediction accuracy of the model. We acquired ERA5 data and GPM data from 2017 to 2020, using the data of the first three years as the training set and the data of the fourth year as the test set. Each sample contains 8 h of data, where the first 4 h of multimodal data are model inputs and the last 4 h of target area GPM precipitation data are model prediction labels.

Before the data are input into the model, we need to do a data transformation on the raw data. First of all, the logarithmic transformation method is used to transform the hourly precipitation value in GPM. The purpose is to enlarge the value with small precipitation to make the data characteristics more obvious, so that the model can better judge whether there is precipitation or not. Second, Z-SCORE transformation is carried out on the multi-modal data of ERA5, a variety of different meteorological data are standardized and data of different magnitudes are converted into unified measured Z-SCORE scores for comparison, which improves the comparability between data and makes it easier for the model to acquire the characteristics of the data of different modes.

The logarithmic transformation formula is:(5)$$y=ln(0.5+10^{2}*x)\;$$
where *x* is the initial value of precipitation data, and *y* is the value after logarithmic transformation. 

The standardized formula of Z-SCORE is:(6)$$z=\frac{x-\mu }{\sigma }$$
where *x* is the initial value of meteorological data, *z* is the value after Z-score standardization, *μ* is the mean value of the overall data, and *σ* is the standard deviation of the overall data. 

In the training model part, the loss function used is the sum of root mean square error (MSE), mean absolute value error (MAE) and dice loss. Among them, MSE and MAE are common loss functions of spatio-temporal prediction models, and the smaller the value, the closer the truth and prediction are. Dice loss is often used in semantic segmentation problems. It is a measure function used to evaluate the similarity of two samples. The value range is between 0 and 1, and the smaller the value, the more similar it is. 

The calculation formula of MSE is:(7)$$MSE=\frac{1}{N}\overset{N}{\underset{i≧1}{\sum }}{\left(y_{i}-{\tilde{y}}_{i}\right)}^{2}$$

The calculation formula of MAE is:(8)$$MAE=\frac{1}{N}\overset{N}{\underset{i≧1}{\sum }}\left|y_{i}-{\tilde{y}}_{i}\right|$$

The calculation formula of dice loss is:(9)$$I=\overset{N}{\underset{i≧1}{\sum }}Y_{i}{\tilde{Y}}_{i}\;U=\overset{N}{\underset{i≧1}{\sum }}\left(Y_{i}+{\tilde{Y}}_{i}\right)\;L_{dice}=1-\frac{2I+\varepsilon }{U+\varepsilon }$$
where *y_i_* is the ground truth value. ${\tilde{y}}_{i}$ is the prediction result. *Y_i_* is the binarization of $y_{i}$ of 0 and 1. ${\tilde{Y}}_{i}$ is the binarization of ${\tilde{y}}_{i}$ of 0 and 1. ε is the smoothing coefficient. 

The complete training process of the model is shown in Figure 3. 

The input data to the model has four branches. The first branch is to encode the three dimensions of month, day and hour of time. Sine and cosine are used to encode each dimension of time into a two-channel three-dimensional array, and a total of six channels of time-coded data $R^{6\times 16\times 16}$ are obtained. The second branch is the GPM precipitation data $R^{2\times 64\times 64}$ of the target area at a certain time and 30 min before that time. Adding precipitation data with an interval of half an hour can make the model learn more spatial dynamic change information. The third branch is the GPM precipitation data $R^{2\times 128\times 128}$ in a large area centered on the target region, and the time is the same as the second branch. The purpose of adding precipitation data in a larger range is to capture the spatial variation information of the surrounding precipitation region. The fourth branch is the large area ERA5 multi-modal meteorological data $R^{35\times 32\times 32}$ centered on the target area at this time. We add ERA5 data to extract spatio-temporal information of a variety of different meteorological data, simulate real meteorological environment changes and improve the accuracy of precipitation prediction. 

We used an Nvidia A40 video card to train the Pred-SF model. The training epoch was set to 20 and the complete training time was about 60 h. After inputting the required historical data, the trained model can get the prediction result in a few seconds.

## 5. Results

### 5.1. Compare Models and Evaluation Indicators

We designed six groups of model comparison experiments to demonstrate the advantages of the Pred-SF model in precipitation prediction and used three common indicators, the critical success index (CSI), probability of detection (POD) and Heidke skill score (HSS), to evaluate the experimental results. The higher the value of these three indexes, the higher the prediction accuracy.

The first group uses the complete Pred-SF model, with input data as multiple meteorological modal data (including time coding) for 4 consecutive hours in a large area and output as the subsequent 4 consecutive hours of precipitation in the target area. The second group used the Pred-SF model without ERA5 data. The input data were the GPM precipitation and time code of the past 4 consecutive hours, and the output was the same as that of the first group. The third group uses Pred, the first part of the Pred-SF model, that is, only the spatial feature coding structure plus the PredRNN-V2 network, with the same input and output as the first group. The fourth to sixth groups used the PredRNN-V2, E3D-LSTM and MIM models, respectively, with the same input and output as the first group.

The calculation formula of CSI is: (10)CSI=TP/(TP+FP+FN),

The calculation formula of POD is: (11)POD=TP/(TP+FP),

The calculation formula of HSS is: (12)HSS=[TP/(TP+FN)] − [FP/(FP+TN)],
where TP represents true positive (actual rain and predicted rain), FP represents false positive (actual no rain but predicted rain), FN represents false negative (actual rain but predicted no rain), and TN represents true negative (actual no rain and predicted rain). 

### 5.2. Qualitative Evaluation

In this part, we first make a quantitative evaluation of the prediction results of various models. Table 2, Table 3 and Table 4 show the ability of six groups of different models to predict different levels of precipitation, and the results are represented by the CSI, POD and HSS indices. It can be seen that the values of each test index of the Pred-SF model are higher than those of the Pred-SF model without ERA5 multimodal data and the Pred model. It can be preliminarily shown that the prediction method combined with multi-modal data to simulate the real meteorological environment and the two-step prediction structure can effectively improve the accuracy of the model to predict the precipitation in the target region. At the same time, the Pred model is slightly higher than the initial PredRNN-V2 model in various values, indicating that the use of a spatial encoding structure can promote multi-modal data fusion and improve the prediction accuracy to a certain extent. Compared with the indexes of E3D-LSTM and MIM, the Pred-SF model can also be proved to have certain advantages in prediction accuracy.

To more effectively analyze the precipitation forecasting ability of the six groups of models, the CSI, POD and HSS indices of each prediction moment of these models were selected and divided into four precipitation levels for display. As can be seen from Figure 4, Figure 5 and Figure 6, all indicators of the six groups of models are gradually decreasing as the time axis goes on, but the index data of the complete Pred-SF model is always the highest, indicating that the model has certain forecasting advantages in each prediction moment of each precipitation grade.

As can be seen from Figure 5a and Figure 6a, although the complete Pred-SF model has the best prediction effect, there is a small gap between the POD index of the Pred model and the Pred-SF model in light rain and above grades (≥0.1 mm/h). However, with the increase in precipitation level, the POD index of the complete Pred-SF model was more and more ahead than that of the Pred model. This indicates that the spatial information fusion network in the second step of the Pred-SF model can effectively capture more spatial information of multi-modal data, so as to improve the prediction ability of the model for large precipitation.

In addition, it can be seen from each line chart that the indexes of the Pred-SF model without ERA5 data decline the fastest over time, which indicates that the method that only uses the historical precipitation region to predict the subsequent precipitation region has poor stability and fails to take the overall climate environment of the region into account, so it is unable to make long-term precipitation prediction.

### 5.3. Visualization and Analysis of Prediction Results

As can be seen from Figure 7 and Figure 8, compared with Pred-SF (no ERA5), Pred, PredRNN-V2, E3D-LSTM and MIM, the precipitation prediction area of each grade of the complete Pred-SF model is close to the ground reality; that is, the precipitation prediction accuracy of each grade is stronger than that of other models. In Figure 7, compared with the complete Pred-SF, the precipitation prediction area of Pred-SF (no ERA5) would dissipate rapidly after the second hour, indicating that the failure to use multi-meteorological mode data fusion would lead to instability of the model and failure to make long-term accurate predictions of subsequent precipitation. At the same time, it can be found from Figure 8 that the yellow precipitation area of the Pred model has a large gap compared with the ground reality; that is, the prediction ability of heavy rain and above (≥8 mm/h) precipitation is weak, which is obviously inferior to the complete Pred-SF model, indicating that the spatial fusion (SF) network is very important to improve the prediction accuracy of the model of large precipitation.

We can draw the following conclusions from the above figures and tables:

First, similar to the Pred-SF (no ERA5) model, in the method of using the historical precipitation region to predict the subsequent precipitation region, compared with the comprehensive prediction using the multi-meteorological mode data, the prediction accuracy drops faster over time. This shows that the multi-modal fusion method can effectively improve the stability and accuracy of the prediction model.

Second, compared with the complete Pred-SF model, only using the Pred model to predict precipitation has a small gap in the prediction accuracy of small rainfall (≥0.1 mm/h), but a large gap in the prediction ability of large rainfall (≥2.5 mm/h and ≥8 mm/h), which proves that the spatial information fusion network in the second part of the Pred-SF model can effectively improve the heavy rain prediction ability of the model.

Third, compared with PredRNN-V2, the Pred model has a small improvement in index values, which indicates that the addition of a spatial coding structure can better capture the correlation between various meteorological data and carry out feature fusion, thus improving the prediction accuracy of the model.

Fourth, compared with PredRNN-V2, E3D-LSTM and MIM, the Pred-SF model has improved index values and prediction effects, which indicates that the model has certain advantages compared with the traditional spatio-temporal prediction model in multi-factor precipitation prediction.

## 6. Conclusions

In this study, we propose a model Pred-SF that is suitable for predicting precipitation in the target region. The focus of this model is the combined self-cyclic prediction method of multi-modal data and the stepwise prediction structure. Pred-SF divides a precipitation prediction problem into two steps. First, the spatial encoding structure and PredRNN-V2 network are used to extract and fuse the multivariate spatio-temporal characteristics of multi-modal data in a large area to make a preliminary prediction of the multi-modal data. Then the spatial information fusion network is used to further integrate the spatial characteristics of the multi-modal data of the output results of the first part and finally output the predicted precipitation value of the target region. The two-step prediction is realized by calculating two loss values, and the deep learning gradient descent algorithm is used to optimize parameters in the process of model training. We used six sets of models to predict the continuous precipitation of the target area for 4 h. Through quantitative evaluation of multiple indicators and the visualization of the results, it can be concluded that the Pred-SF model has a good ability to predict the precipitation of the target region.

Of course, there is still a lot of room for improvement in the current Pred-SF. The deep learning model can be used to predict precipitation in the target area, which can be improved mainly from two aspects: model structure and data.

Pred-SF uses a stepwise prediction structure, which can be improved both in the spatial encoding structure and PredRNN-V2 in the first part and in the spatial information fusion structure. The PredRNN-V2 we selected only uses the convolution structure. Currently, Transformer is challenging the position of convolution in the field of machine vision, such as VIT [26] and Swin Transformer [27]. How to use the attention mechanism to improve the model’s ability to extract and fuse multivariate data features is one of the future directions of Pred-SF.

In this study, we used two kinds of data to verify the validity of the model; the first is GPM global precipitation measurements and the second is ERA5 global climate atmospheric reanalysis data. For the precipitation prediction problem, these data cannot represent the complete and absolutely real changes in the precipitation weather system. To improve the accuracy of forecast and to be used in specific operations, it is necessary to generate high-quality meteorological data in real time. For Pred-SF, measured radar data or measured satellite data can be added to simulate a more real climate environment and provide more learnable spatio-temporal information of weather systems for the training of precipitation models.

## Figures and Tables

**Figure 1 sensors-23-02609-f001:**
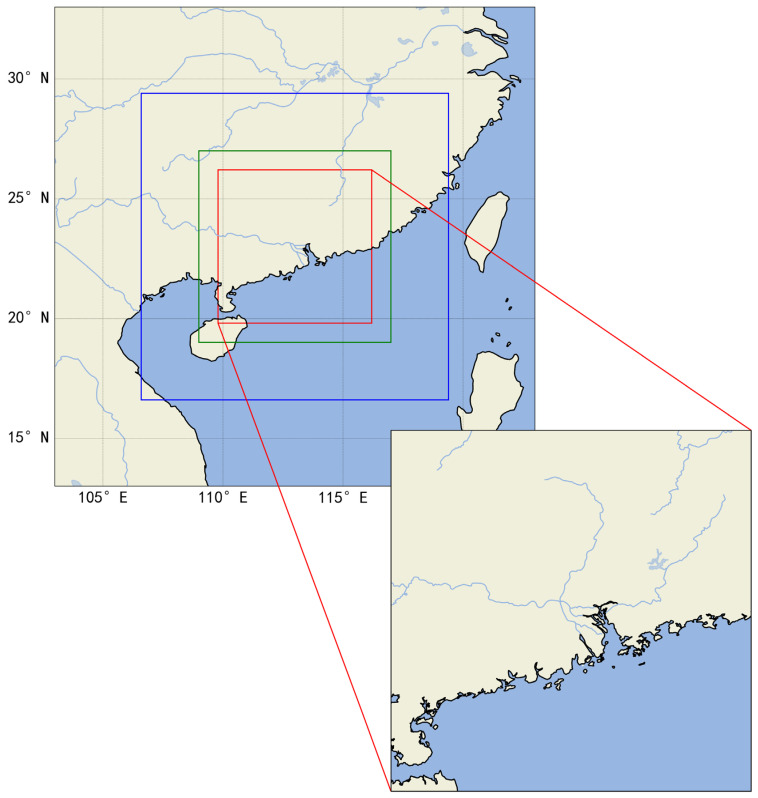
Visualization of the study area. The red box is the research area of this paper, the blue box is the large-scale GPM precipitation data input area, and the green box is the ERA5 multi-meteorological factor data input area.

**Figure 2 sensors-23-02609-f002:**
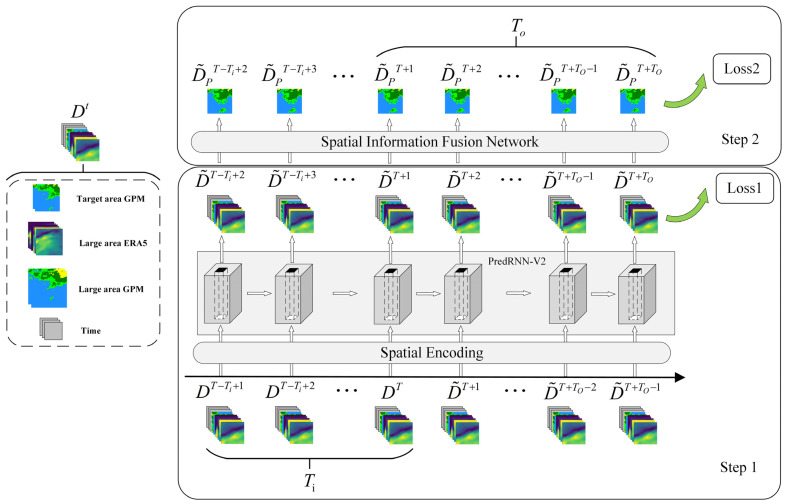
The Pred-SF model is divided into two steps. The first step is the spatial encoding structure and PredRNN-V2 network (Pred), which input multiple meteorological modal data $D^{t}$ including precipitation and output multiple meteorological modal data predicted value ${\tilde{D}}^{t+1}$. The second step is the spatial information fusion network (SF), which fuses the spatial features of the predicted value ${\tilde{D}}^{t+1}$ and outputs the predicted precipitation value ${\tilde{D}}_{P}{}^{t+1}$ in the target region. To make the Pred-SF run step by step, we calculated two losses for the front and back parts, so that the model could optimize the parameters according to the requirements of the model in the training process.

**Figure 3 sensors-23-02609-f003:**
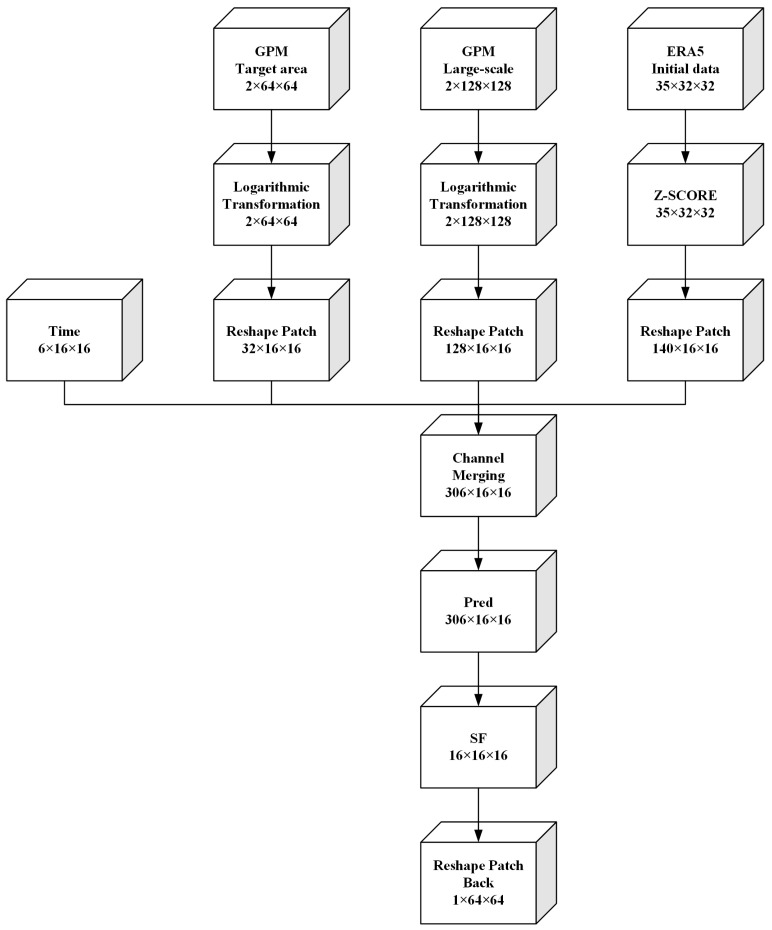
The training process of Pred-SF shows the changes of C × H × W dimensions of data in the whole process from input to output.

**Figure 4 sensors-23-02609-f004:**
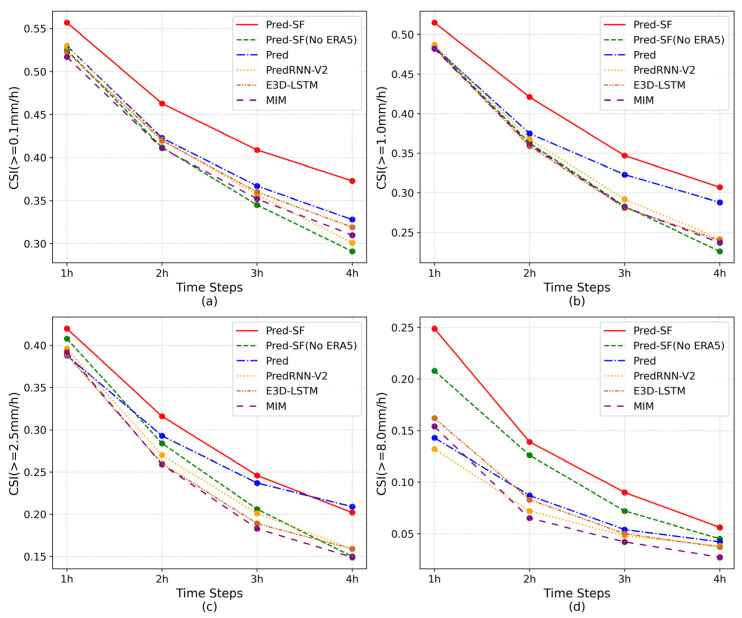
CSI index of the six groups of models at each prediction moment: (**a**) precipitation is greater than 0.1 mm/h; (**b**) precipitation is greater than 1.0 mm/h; (**c**) precipitation is greater than 2.5 mm/h; (**d**) precipitation is greater than 8.0 mm/h.

**Figure 5 sensors-23-02609-f005:**
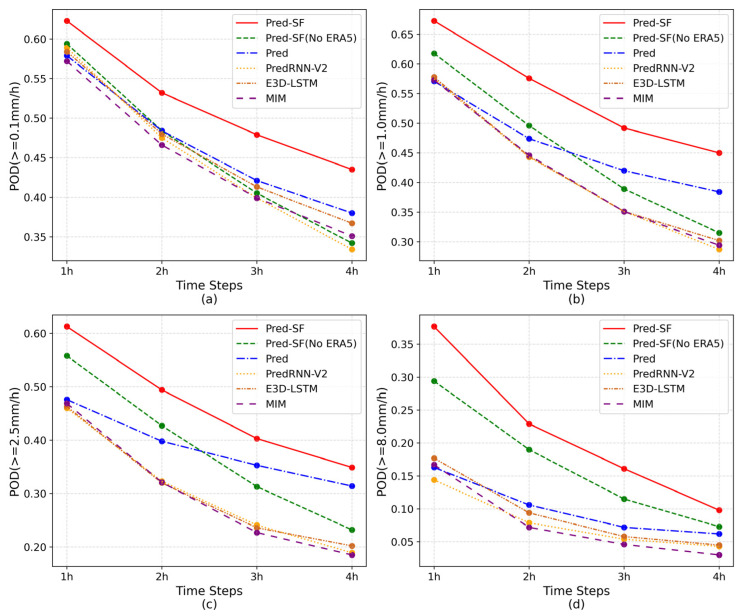
POD index of the six groups of models at each prediction moment: (**a**) precipitation is greater than 0.1 mm/h; (**b**) precipitation is greater than 1.0 mm/h; (**c**) precipitation is greater than 2.5 mm/h; (**d**) precipitation is greater than 8.0 mm/h.

**Figure 6 sensors-23-02609-f006:**
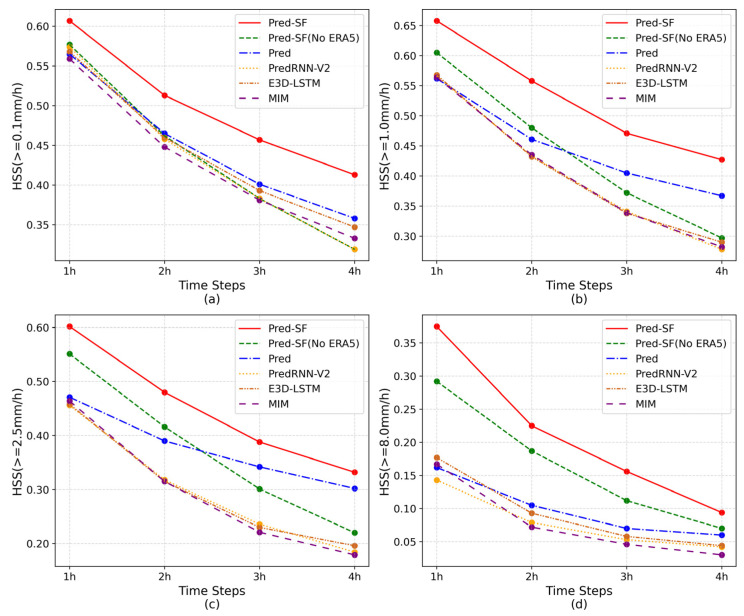
HSS index of the six groups of models at each prediction moment: (**a**) precipitation is greater than 0.1 mm/h; (**b**) precipitation is greater than 1.0 mm/h; (**c**) precipitation is greater than 2.5 mm/h; (**d**) precipitation is greater than 8.0 mm/h.

**Figure 7 sensors-23-02609-f007:**
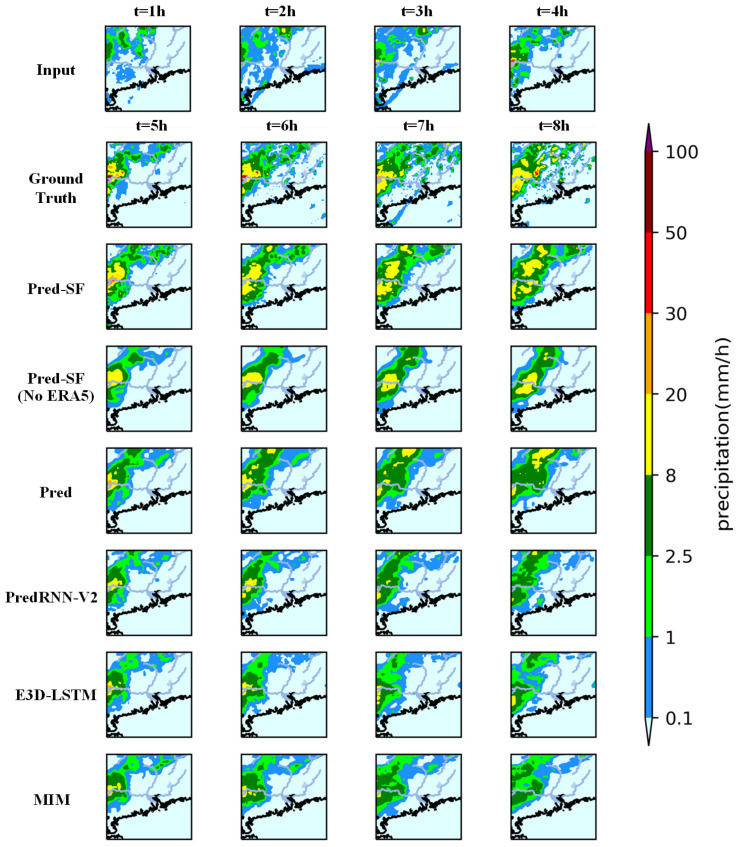
The precipitation prediction results of the six groups of models for 4 consecutive hours (4 time steps) are shown in the figure. The first row is the input, the second row is the ground truth and the following three rows are the output results of the three groups of models.

**Figure 8 sensors-23-02609-f008:**
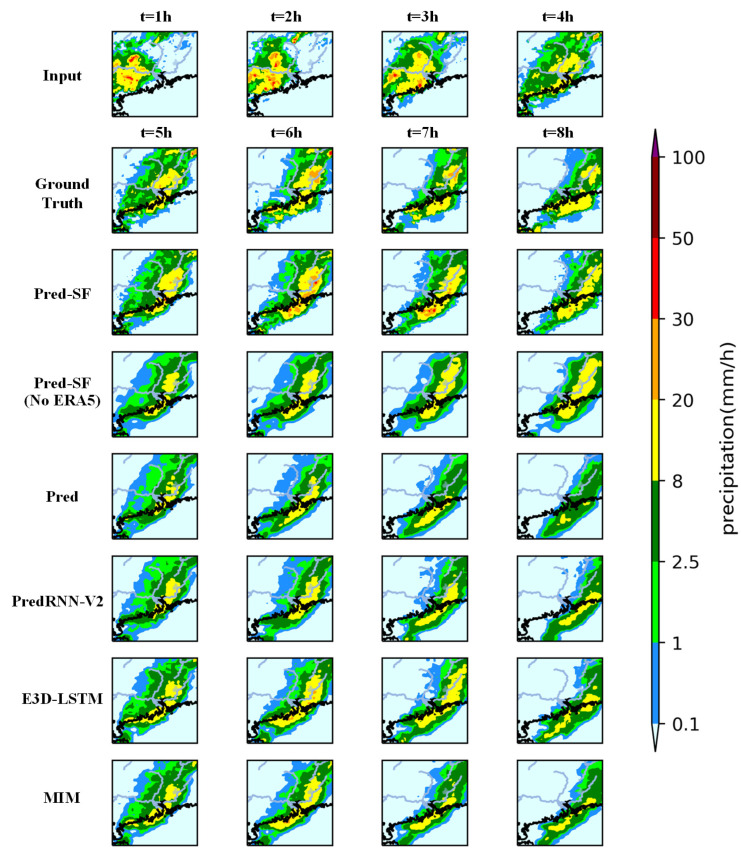
The precipitation prediction results of the six groups of models for 4 consecutive hours (4 time steps) are shown in the figure. Same as Figure 7.

**Table 1 sensors-23-02609-t001:** Data information.

Data Type	Specific	Vertical Level	Spatial Resolution	Temporal Resolution
GPM	Precipitation	Surface	0.1° × 0.1°	0.5 h
ERA5	Temperature U wind V wind Vertical wind Specific humidity Cloud cover Percentage of cloud liquid water content	1000 hpa 950 hpa 850 hpa 700 hpa 500 hpa	0.25° × 0.25°	1 h

**Table 2 sensors-23-02609-t002:** CSI indicators of three groups of different models. We set three different levels: light rain and above (≥0.1 mm/h), moderate rain and above (≥2.5 mm/h) and heavy rain and above (≥8 mm/h). CSI and POD indexes were calculated for the three grades.

Model	CSI (4 h Average)		
	≥0.1 mm/h	≥2.5 mm/h	≥8 mm/h
Pred-SF	0.451	0.297	0.134
Pred-SF (No ERA5)	0.393	0.262	0.113
Pred	0.411	0.282	0.082
PredRNN-V2	0.402	0.256	0.072
E3D-LSTM	0.406	0.249	0.083
MIM	0.398	0.246	0.073

**Table 3 sensors-23-02609-t003:** POD indicators of three groups of different models.

Model	POD (4 h Average)		
	≥0.1 mm/h	≥2.5 mm/h	≥8 mm/h
Pred-SF	0.518	0.465	0.216
Pred-SF (No ERA5)	0.456	0.382	0.168
Pred	0.466	0.386	0.101
PredRNN-V2	0.449	0.303	0.080
E3D-LSTM	0.461	0.305	0.094
MIM	0.447	0.301	0.079

**Table 4 sensors-23-02609-t004:** HSS indicators of three groups of different models.

Model	HSS (4 h Average)		
	≥0.1 mm/h	≥2.5 mm/h	≥8 mm/h
Pred-SF	0.498	0.451	0.213
Pred-SF (No ERA5)	0.435	0.372	0.165
Pred	0.448	0.376	0.099
PredRNN-V2	0.433	0.299	0.079
E3D-LSTM	0.442	0.300	0.093
MIM	0.430	0.295	0.079

## Data Availability

GPM data can be obtained from the website https://disc.gsfc.nasa.gov/ (accessed on 30 June 2021); ERA5 data can be obtained from the website https://cds.climate.copernicus.eu/ (accessed on 10 March 10 2022).

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
