# Peer review of "Pred-SF: A Precipitation Prediction Model Based on Deep Neural Networks"

_sensors, 2023, doi:10.3390/s23052609_

Round 1

Reviewer 1 Report

This paper presents Pred-SF model for precipitation prediction Based on Deep Neural Networks in target areas. This paper is organized not bad but the following comments need to be considered in the revision

1. In this paper, the Prediction Model Based on Deep Neural Networks is considered, how to show the novelty of this method? Moreover, any other Neural Networks approaches can be considered and compared, such as: Adaptive Optimal Control for a Class of Nonlinear Systems: The Online Policy Iteration Approach; Online policy iterative-based H∞ optimization algorithm for a class of nonlinear systems.

2. In general, the authors should give the main algorithm procedure of the designed methods;

3. The authors are encouraged to add the comparisons in the paper, especially in the simulation parts.

4. There exist language mistakes and typos throughout the paper, and authors should check the paper carefully.

Reviewer 2 Report

The authors propose a model for precipitation prediction in target areas called Pred-SF. The model combines multiple meteorological modal data to carry out the self-cyclic prediction and strip prediction as a structure. The target of precipitation prediction is validated in collected data trying to anticipate 4 hours ahead.

The topic is interesting, considering the extrapolation of using more than one source beyond sensors to collect data, and it can be helpful.

Considering the data dynamicity, it is not clear how long time is spent on retraining and re-running the model. The authors should show this metric in the results.

The authors show results near 50% to correctly predict one hour ahead and near 0 once considering four hours ahead, not justifying the use of the model. It is necessary to keep clear the benefits of using the model even with these results.

The first three paragraphs in Section 2.2 can be merged into one. Lines 133-147

An example of the dataset columns and values should be shown for the readers.

Reviewer 3 Report

This is a fascinating and practical study. The proposed model and its outcomes are significant and compelling. However, the authors should consider the following suggestions: 

The authors emphasize the effectiveness of their model in predicting the subsequent 4-hour continuous precipitation in a target area in their manuscript. However, the authors should explain why they chose such a setting and why it makes sense. 

Authors should include other studies or explanations about different types of n-hour continuous precipitation before deciding to use 4-hour continuous precipitation in a target area. It would be helpful to know which four hours the authors are utilizing. Will the range affect the proposed method's outcome? 

The authors should demonstrate how they keep their tests from overfitting or underfitting problems. 

Instead of comparing Pred and its variance in the results section, the authors should compare their method to the existing one.

Round 2

Reviewer 2 Report

The authors updated the paper, and I now understand it as sufficient for publication.